# Added Value of CCTA-Derived Features to Predict MACEs in Stable Patients Undergoing Coronary Computed Tomography

**DOI:** 10.3390/diagnostics12061446

**Published:** 2022-06-12

**Authors:** Valeria Pergola, Giulio Cabrelle, Giulia Mattesi, Simone Cattarin, Antonio Furlan, Carlo Maria Dellino, Saverio Continisio, Carolina Montonati, Adelaide Giorgino, Chiara Giraudo, Loira Leoni, Riccardo Bariani, Giulio Barbiero, Barbara Bauce, Donato Mele, Martina Perazzolo Marra, Giorgio De Conti, Sabino Iliceto, Raffaella Motta

**Affiliations:** 1Cardiology Unit, Department of Cardio-Thoraco-Vascular Sciences and Public Health, University of Padua, 31100 Padua, Italy; g.mattesi17@gmail.com (G.M.); lillo.dellino79@gmail.com (C.M.D.); saveriocontinisio@gmail.com (S.C.); carolina.montonati@gmail.com (C.M.); loira.leoni@aopd.veneto.it (L.L.); riccardo.bariani@unipd.it (R.B.); barbara.bauce@unipd.it (B.B.); donato.mele@unipd.it (D.M.); martina.perazzolomarra@unipd.it (M.P.M.); sabino.iliceto@unipd.it (S.I.); 2Department of Medicine—DIMED, University of Padova, 31100 Padua, Italy; giulio.cabrelle@gmail.com (G.C.); adelaide.giorgino@gmail.com (A.G.); chiara.giraudo@unipd.it (C.G.); raffaella.motta@unipd.it (R.M.); 3Radiology Unit, Conegliano Hospital, 31100 Treviso, Italy; simocatta91@hotmail.it; 4Radiology Unit, Castelfranco Hospital, 31100 Treviso, Italy; antonio.furlan@gmail.com; 5Radiology Unit, Padua University Hospital, 31100 Padua, Italy; giulio.barbiero@aopd.veneto.it (G.B.); giorgio.deconti@aopd.veneto.it (G.D.C.)

**Keywords:** coronary computed tomography angiography, pericoronary fat attenuation index, low attenuation plaque, major adverse cardiac events, plaque analysis

## Abstract

Clinical evidence has emphasized the importance of coronary plaques’ characteristics, rather than lumen stenosis, for the outcome of cardiovascular events. Coronary computed tomographic angiography (CCTA) has a well-established role as a non-invasive tool for assessing plaques. The aim of this study was to compare clinical characteristics and CCTA-derived information of stable patients with non-severe plaques in predicting major adverse cardiac events (MACEs) during follow-up. We retrospectively selected 371 patients (64% male) who underwent CCTA in our center from March 2016 to January 2021 with Coronary Artery Disease—Reporting and Data System (CAD-RADS) 0 to 3. Of those, 198 patients (53% male) had CAD-RADS 0 to 1. Among them, 183 (49%) had normal pericoronary fat attenuation index (pFAI), while 15 (60% male) had pFAI ≥ 70.1 Hounsfield unit (HU). The remaining 173 patients (76% male) had CAD-RADS 2 to 3 and were divided into patients with at least one low attenuation plaque (LAP) and patients without LAPs (n-LAP). Compared to n-LAP, patients with LAPs had higher pFAI (*p* = 0.005) and had more plaques than patients with n-LAP. Presence of LAPs was significantly higher in elderly (*p* < 0.001), males (*p* < 0.001) and patients with traditional risk factors (hypertension *p* = 0.0001, hyperlipemia *p* = 0.0003, smoking *p* = 0.0003, diabetes *p* = <0.0001, familiarity *p* = 0.0007). Among patients with CAD-RADS 0 to 1, the ones with pFAI ≥ 70.1 HU were more often hyperlipidemic (*p* = 0.05) and smokers (*p* = 0.007). Follow-up (25,4 months, range: 17.6–39.2 months) demonstrated that LAP and pFAI ≥ 70.1 significantly and independently (*p* = 0.04) predisposed to outcomes (overall mortality and interventional procedures). There is an added value of CCTA-derived features in stratifying cardiovascular risk in low- to intermediate-risk patients with non-severe, non-calcified coronary plaques. This is of utmost clinical relevance as it is possible to identify a subset of patients with increased risk who need strengthening in therapeutic management and closer follow-up even in the absence of severe CAD. Further studies are needed to evaluate the effect of medical treatments on pericoronary inflammation and plaque composition.

## 1. Introduction

Clinical evidence has emphasized the importance of coronary plaque characteristics, rather than lumen stenosis severity, for the outcome of cardiovascular events [1,2,3,4,5,6,7,8]. Coronary computed tomography angiography (CCTA) represents an easily accessible diagnostic technique to assess plaques, and it is gaining strength for a stronger risk stratification of patients affected by coronary artery disease (CAD).

The Coronary Artery Disease—Reporting and Data System (CAD-RADS) was firstly proposed in 2016 as a standardized method to describe coronary findings in CCTA [2]. Four large, randomized trials (CT-STAT, ACRIN-PA, ROMICAT II and CT-COMPARE) systematically proved the safety of a negative coronary CCTA in the acute chest pain setting to identify patients for discharge from the emergency department [3,4,5,6,7].

CCTA also permits a detailed evaluation of the morphology and composition of plaques [8]. Different cut-offs expressed as Hounsfield unit (HU) are specific for the various components of the coronary plaque, and their quantification can discriminate between low attenuation soft plaques (LAPs), high attenuation plaques (HAPs) and calcific plaques [9]. This kind of analysis overcomes the limitation of determining coronary calcium by the “Agatston score” [10], as it not only quantifies the proportion and distribution of calcium, but it also adds qualitative and quantitative information about fibrous and adipose components of the plaques [11]. It is also well established that highly calcific plaques are more stable compared to LAPs [11]. Moreover, the CCTA high-risk coronary plaque characteristics were shown to predict acute coronary syndrome (ACS) among patients with acute chest pain [6,8].

There is also increasing evidence that pericoronary fat inflammation (pFAI) measured in the right coronary artery (RCA) is an independent predictor of cardiovascular events and mortality [12,13,14,15,16,17]. In addition, other evidence exists about the role of elevated pFAI in patients with myocarditis, myocardial infarction with nonobstructive coronary arteries (MINOCA) and Takotsubo syndrome [13,14]. Indeed, pericoronary adipose tissue releases proinflammatory cytokines that can alter the structure of the coronary wall, leading to CAD [15]. Previous studies indicated that pFAI correlates with plaque attenuation components in patients with non-ST-elevation acute coronary syndrome [16]. However, studies based on the relationship between pFAI, plaque composition and outcomes in stable patients with non-obstructive CAD are scarce.

The aim of this study was to compare clinical characteristics and CTCA-derived information of patients with mild to moderate plaques without acute chest pain, in predicting major adverse cardiac events (MACEs) during follow-up. 

## 2. Materials and Methods

### 2.1. Study Population

Data of patients who underwent CCTA from 1 March 2016 to 31 June 2021 in Padua University Hospital were retrospectively retrieved. According to guidelines [18], in our department, CCTA is requested for evaluation of asymptomatic patients or patients with stable (typical or atypical) chest pain, with a low to intermediate pre-test probability of obstructive CAD. We excluded patients with the following characteristics: age < 18 years; acute chest pain or ECG changes suggestive of ACS; CAD-RADS > 3; blooming artifacts for wide coronary calcification; history of previous cardiac surgery, coronary dissection, congenital heart disease, coronary origin anomaly; active coexisting inflammatory or oncological disease. Moreover, according to current practice [2] and to our internal policy, patients classified as CAD-RADS 3 are always invited to perform a physical or pharmacological stress test and were excluded in case of percutaneous or surgical procedures performed within 3 months from a pathological stress test. Four hundred forty patients were evaluated; of those, 69 were excluded because of poor quality images and or lack of follow-up data. Our final population consisted of 371 patients (237 male); of those, 173 patients (76% male) had CAD-RADS 2 to 3, while 198 patients (53% male) had CAD-RADS 0 to 1 and served as control group. Among patients with CAD-RADS 2 to 3, the ones with at least 1 LAP were considered in the LAP group, while the others were considered as non-LAPs (n-LAPs). Two expert physicians (1 radiologist and 1 cardiologist) retrieved the medical records of all patients. Each patient’s clinical history, medications, echocardiographic parameters and laboratory data (high-sensitivity troponin (TnI), glomerular filtration rate (eGFR)) were collected. 

### 2.2. CCTA Technique

CCTA was performed using a 320 × 0.5 mm MDCT (Toshiba Aquilion ONE ViSION Edition; Canon Medical Systems Corporation, Ōtawara, Tochigi, Japan). Our acquisition prospective protocol was previously described [19,20]. Data were transferred to an external workstation (Vitrea2 FX version 6.3, Vital Images, Plymouth, MN, USA) providing an automatic quantitative assessment of plaque constitution and stenosis severity. The plaques were assessed using original axial images, multiplanar reformation and cross-sectional reconstruction. Coronary plaque characteristics were analyzed across each segment of the main coronary arteries; if necessary, manual editing of the centerline and lumen contours was performed in order to improve the accuracy of plaque volume and stenosis quantification. Coronary arteries were defined as normal if no plaques were visually detectable. Percentage of stenosis was calculated by the following formula: minimum luminal area/mean luminal area at proximal and distal reference. The normal vascular diameter of the proximal point was set as 100%. The degree of stenosis was divided into the following grades according to the percentage of vascular reduction: Grade 0: normal, no coronary artery stenosis; Grade 1: stenosis < 25%, minimal stenosis, often manifested as an irregular lumen; Grade 2: stenosis of 25% to 50%, mild stenosis; Grade 3: stenosis of 50% to 70%, moderate stenosis. As for diffused coronary lesions, we examined the segment from the proximal normal segment to the distal normal segment across the lesion. According to the literature [6,19], the plaques with total radiological density <30 HU and a volume > 15 mm^3^ or those with density between 30 and 60 HU and a volume > 52 mm^3^ were considered as hypodense or LAPs (Figure 1). 

The plaques with total radiological density > 60 HU were considered as non-LAPs (n-LAPs). Volumetric characterization of the plaque characteristics focused on the entire plaque volume under 3-dimensional (3D) reconstruction, and the cross-sectional characterization focused on the level of the minimal lumen area. When a CT study presented separate lesions, we considered in the analysis the one with the lowest density, independently from the grade of stenosis; therefore, the patient was classified in the LAP group if a plaque with LAP characteristics was present. In addition to low radiological density, other vulnerable plaque features (positive remodeling, defined as remodeling index >1.1; spotty calcification (SC), defined as intraplaque calcification ≤3 mm; and napkin-ring sign, defined as low intraplaque attenuation surrounded by a high attenuation rim) were reported. 

### 2.3. Pericoronary Fat Attenuation

As described by Antonopoulos et al., pFAI was calculated in the right coronary artery (RCA) [21] using Aquarius Workstation (TeraRecon Inc., Foster City, CA, USA) in a 40 mm long region of interest (ROI) starting 10 mm away from the vessel origin. Adipose tissue volume and median and mean HU within the range [−190; −30] were automatically calculated, as described previously by our group [12] (Figure 1). To overcome observer bias, our pFAI physicians’ panel (1 cardiologist, 1 radiologist) was not involved in obtaining plaque analysis images.

### 2.4. Clinical Endpoint

The outcome panel was formed by two physicians blinded to the CCTA findings. A MACE was defined as hospital admissions for any kind of ACS or a positive functional diagnostic testing leading to percutaneous or surgical intervention [6]. Cardiovascular-related death was defined as death from myocardial infarction, aggravation of heart failure, sudden cardiac death or ischemic stroke. If a patient had two or more clinical events, only the first event was included as an outcome. Follow-up was ended on 22 February 2022 or whenever a primary outcome event occurred. 

### 2.5. Statistical Analysis

Data are presented as the means ± standard deviation (SD) and relative frequencies (%) when normally distributed or as the medians and interquartile range (IQR) when not normally distributed for continuous variables and as percentages for categorical variables. Univariable analysis was performed by χ^2^ test for categorical variables and by ANOVA with Dunnett post hoc test for continuous variables. To identify the independent predictors of outcome, two multivariable analyses (logistic regression) were performed (for mortality and for composite outcome, respectively). Only the variables that emerged as significant at univariate analysis were included in the multivariable models. The confidence interval (CI) was set at 95% with *p* < 0.05. A receiver operating characteristic (ROC) curve showing the relationship between clinical sensitivity and specificity for every possible cut-off of pFAI was constructed. Statistical analysis was performed using IBM SPSS Statistics for Windows software (Version 25.0. Armonk, NY, USA: IBM Corp. 2017).

## 3. Results

### 3.1. Population Characteristics

Three hundred seventy-one patients (237 male, mean age 55.5 ± 11.9 years) were included; of those, 173 patients (76% male, mean age 62.5 ± 11.9 years) had CAD-RADS 2 to 3, while 198 patients (53% male, mean age 50.7 ± 14.9 years) had CAD-RADS 0 to 1 and served as a control group. Among the latter, 15 patients (7% of CAD-RADS 0-1, 60% male, mean age 56,3 ± 15,5 years) had pFAI ≥ −70.1 HU. There were 136 patients with at least one LAP (37%, 65.3 ± 10.4 years), *while 37 patients had n-LAPs (10%, 59.5 ± 13.5 years, p = 0.005)*. Table 1 describes the clinical and laboratory characteristics of the population. LAPs were more frequent in elderly (*p* < 0.001) and in patients with traditional cardiovascular risk factors, such as hypertension (*p* < 0.001), diabetes mellitus (*p* < 0.001), smoking (*p* < 0.001), CAD familiarity (*p* < 0.001) and hyperlipemia (*p* = 0.05), compared to n-LAPs. Patients with LAPs had a slightly lower eGFR than those without (*p* < 0.036). 

Interestingly we found among patients with CAD-RADS ≤1 and pFAI ≥ −70 HU, despite their sparse number (N = 15), a higher prevalence of smoking (*p* = 0.0079) and hyperlipidemia (*p* = 0.05) compared to patients with CAD-RADS ≤ 1 and lower pFAI. Compared to patients with LAPs, the clinical characteristics of patients with CAD-RADS ≤ 1 and pFAI ≥ −70 HU were similar, except for age (*p* = 0.003 vs. LAP).

Compared to patients without plaques, the overall population of patients with plaques took more antiplatelets and anticoagulants, together with antihypertensives (angiotensin-converting enzyme (ACE) inhibitors), lipid-lowering agents (statins) and diuretics (Table 2). Patients with pFAI ≥ −70 HU took more anticoagulants and Ca antagonists (*p* < 0.001) compared to normal controls. 

### 3.2. CCTA Features and pFAI Values 

A total of 370 plaques from 173 patients were analyzed; patients with LAPs had more plaques per coronary than subjects with n-LAPs (321 vs. 49, *p* < 0.001) (Table 3). The overall volume of LAPs was greater than that of n-LAPs in all the main coronaries (*p* < 0.001) (Table 3). The presence of other plaque instability signs (napkin-ring sign, irregular plaque margins, positive remodeling and spotty calcifications) was significantly higher in the LAP group (*p* = 0.0208). In addition, the mean pFAI was higher in the LAP group compared with n-LAP (*p* = 0.0159). 

### 3.3. Outcome Analysis

Median follow-up was 25.4 months (IQR = 17.6/39.2 months). Presence of LAPs (*p* = 0.04) and high-risk plaque signs (*p* = 0.03) independently predisposed to MACEs as a composite outcome (mortality, hospitalization for ACS) (Table 4a). When the outcomes of the whole population were analyzed according to pFAI −70 HU as cut-off, the overall cardiovascular mortality and the number of percutaneous coronary angioplasty/stenting were significantly higher in patients with mean pFAI ≥ −70 HU. The number of percutaneous coronary angioplasty/stenting was higher (*p* = 0.05) in patients with mean pFAI ≥ −70 HU (Table 4b).

LAP significantly and independently predisposes to composite outcome (*p* = 0.0202) and cardiovascular mortality (*p* = 0.0152), even when corrected for traditional risk factors. PFAI significantly and independently predisposes to cardiovascular mortality, even when corrected for traditional risk factors (*p* = 0.0015) (Table 5). The ROC curve showed an excellent accuracy for pFAI in predicting MACEs (Figure 2).

## 4. Discussion

Due to the relatively high occurrence and severity of events in patients with non-obstructive CAD [13,14,15,16], the incorporation of CCTA into clinical decision-making can be of great importance for improved identification of adverse events. The main finding of our study is that there is a positive correlation between CTCA-derived features and MACEs in our very selected population of patients with mild to moderate CAD. Our results demonstrated that integration of plaque analysis and pFAI into clinical risk evaluation has the potential to better stratify cardiac risk over anatomical coronary artery disease degree, even in plaques not angiographically significant. The novelty of our study compared to previous studies is that we analyzed CTCA-derived features in a non-acute setting.

The diagnostic impact of CCTA in suspected ACS has been extensively documented [3,4,5,6,7,8]. Due to its high negative predictive value, its diagnostic usefulness was first demonstrated in patients presenting to the emergency department with acute chest pain in large clinical trials [3,4,5,6,7]. Recently, the prognostic value of CCTA in non-ST-elevation acute coronary syndromes was also demonstrated [22]. Our results are in agreement with those of Nance et al. [23], who analyzed 458 patients with low to intermediate risk of CAD who presented to the emergency room with ACP and had negative ECG. After adjustment for clinical characteristics and Ca-score, they found a higher hazard ratio for non-calcified and partially calcified plaques compared to solely calcified plaques (57.64, 55.76 and 26.45, respectively). Our results are also in accordance with those of Nadjiri et al. [24], who observed, during a clinical follow-up of 5.7 years, that MACEs were associated with all plaque characteristics; in particular, the strongest association was observed for low attenuation plaques (HR 1.12, *p* < 0.0001).

It has been widely demonstrated that pFAI correlates with plaque attenuation components in patients with non-ST-elevation acute coronary syndrome. Antonopoulos et al. [21] firstly reported that the detection and quantification of coronary inflammation is a useful noninvasive imaging biomarker of perivascular inflammation and immune activation which are known to be important determinants of plaque vulnerability Our results are in agreement with those of Gaibazzi et al. [14], who found higher pFAI values in patients with non-obstructive coronary arteries. Interestingly, in accordance with Gaibazzi et al. [14], we also found in our small population of 15 patients with CAD-RADS ≤ 1 and pFAI ≥ −70 HU higher rates of hypertension and smoke habit. With the exception of age and diabetes, the rates of cardiovascular risk factors in our small subgroup were similar to the LAP group, suggesting that cardiovascular risk factors may alter pericoronary fat, leading with time to unfavorable plaques. As highlighted by the CRISP-CT study [12] and the SCOT-HEART study [25], there is a positive correlation between low attenuation plaques, pFAI and cardiovascular events considering all extent of coronary artery disease. Goller et al. [26]. demonstrated that the pFAI was significantly higher around culprit lesions compared to non-culprit lesions in acute coronary syndromes. Sun et al. [17] showed a positive correlation between pFAI and vulnerable plaque characteristics in patients with non-ST-elevation myocardial infarction. Our results are in agreement with those of Yan et al. [27], who found that the combination of pFAI and plaque assessment improved the discrimination of ischemia compared with stenosis assessment alone. However, there are no studies evaluating the correlation between pFAI and plaque composition and outcomes in the specific population of stable patients with non-obstructive CAD. Our study, to the best of our knowledge, is the first one that correlated unfavorable plaque characteristics and pFAI with cardiovascular events in low to intermediate cardiovascular risk patients.

### Limitations

This study has several limitations. First, this study was retrospectively designed. Second, the entire study cohort was relatively small (n = 371). Moreover, the number of the population was scarce, especially for pFAI patients with >−70 HU. To attenuate these limits, we selected the patients using strict inclusion and exclusion criteria to identify the exact effects of risk factors. Third, a relatively short follow-up was considered. Probably because of the shortness of our follow-up, no significant difference was found when the different outcomes were considered separately according to plaque composition, but pFAI alone was able to identify all the outcomes separately. The dosage and the date of start of the medications were not available in all patients; therefore, it was not possible to evaluate any effect on the plaque analysis or pFAI. Further studies are needed to evaluate the effect of medical treatments on pericoronary inflammation and plaque composition.

## 5. Conclusions

CCTA-derived features offer higher prognostic insights than a mere angiographic evaluation or calcium-score analysis, as they are more and more being considered reliable predictors of MACEs. Our study adds evidence on the prognostic role of the CCTA in stable patients with non-obstructive, non-calcific CAD. This means that in patients with mild to moderate stenosis, unfavorable plaque characteristics (LAPs and elevated pFAI) should be considered as predictors of higher risk of cardiovascular events. This information obtained by CCTA is of utmost importance in the clinical scenario to identify patients who need strengthening in medical treatment and a closer follow-up. We could also hypothesize that in the future, CTCA-derived features and in particular pFAI may help to select patients for new therapies targeting coronary inflammation.

## Figures and Tables

**Figure 1 diagnostics-12-01446-f001:**
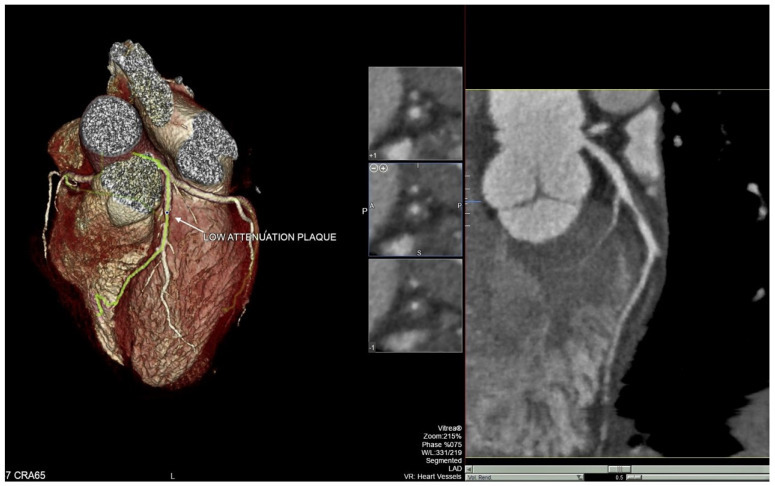
Example of low attenuation plaque in the first tract of the left anterior descending artery (LAD).

**Figure 2 diagnostics-12-01446-f002:**
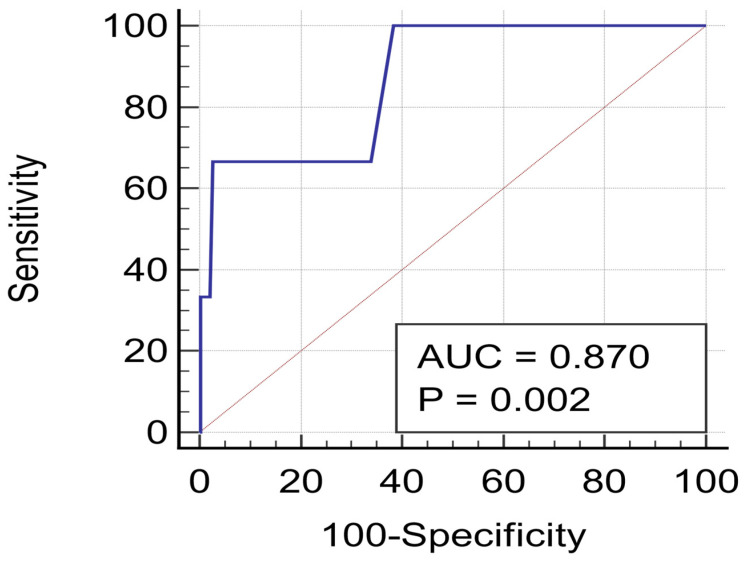
ROC curve for pericoronary fat inflammation in predicting cardiovascular mortality.

**Table 1 diagnostics-12-01446-t001:** Clinical and laboratory characteristics of the population.

Group N (%)	LAP 136 (37)	n-LAP 37 (10)	*p*	CAD-RADS ≤ 1 183 (49)	CAD-RADS < 1 with pFAI ≥ −70 HU 15 (4)	*p*
**Male**	106 (77.9)	25 (67.5)	0.168	90 (49%)	9 (60%)	0.413
**Age (years)**	65.3 ± 10.4	59.5 ± 13.5	**0.005**	52 ± 14.8	56.3 ± 15.5	0.282
**BMI (Kg/m^2^)**	30.2 ± 9.6	26.1 ± 8.2	**0.018**	25 ± 6.9	27.3 ± 4.8	0.207
**eGFR(mL/min/m^2^)**	80.0 ± 20.5	87.9 ± 17.4	**0.036**	93.0 ± 19.2	93.6 ± 21.5	0.908
**Troponin (ng/L)**	14.3 ± 8.7	16.5 ± 5.6	**0.147**	13.9 ± 7.4	13.4 ± 9.1	0.999
**Sinus rhythm**	116 (85.3)	34 (91.8)	0.270	179 (97.8)	15 (100)	0.562
**Hypertension**	100 (49.5)	28 (13.9)	**0.0001**	68 (37.1)	9 (60)	0.079
**Hyperlipemia**	66 (49.6)	22 (16.5)	**0.0003**	45 (24.6)	7 (47.7)	**0.05**
**Smoking**	50 (41.3)	11 (9.1)	**0.0003**	50 (27.3)	9 (60)	**0.007**
**Diabetes**	27 (79.4)	3 (8.8)	**<0.0001**	5 (2.7)	0 (0)	0.520
**CAD familiarity**	50 (42)	14 (11.8)	**0.0007**	60 (32.8)	5 (33.3)	0.981
**Ejection fraction (%)**	57.0 ± 10.1	60.0 ± 6.9	0.0909	58 ± 9.2		

**Legend:** BMI: body mass index; BNP: brain natriuretic peptide; BSA: body surface area; HS-CRP: high-sensitivity C-reactive protein; HAP: high attenuation plaque; LAP: low attenuation plaque; eGFR; glomerular filtration rate; CAD: coronary artery disease.

**Table 2 diagnostics-12-01446-t002:** Patients’ therapy at the time of CCTA.

Group N (%)	LAP 136 (37)	n-LAP 37 (10)	*p*	CAD-RADS ≤ 1 183 (49)	CAD-RADS < 1 with pFAI ≥ −70 HU 15 (4)	*p*
** *ASA* **	55 (40.7)	15 (40.5)	0.982	41 (22.4)	0 (0)	**0.04**
** *Clopidogrel* **	10 (7.4)	1 (2.7)	0.301	2 (1.1)	0 (0)	0.697
** *Coumadin* **	4 (3.0)	1 (2.7)	0.923	2 (1.1)	2 (13.3)	**0.0009**
** *NAO* **	10 (7.5)	1 (2.7)	0.294	4 (2.2)	8 (53.3)	**<0.0001**
** *β-Blockers* **	58 (43)	14 (37.8)	0.585	48 (26.2)	1 (6.7)	0.088
** *Ca Antagonists* **	26 (19.3)	8 (21.6)	0.684	22 (12.0)	7 (46.7)	**0.0002**
** *ACE Inhibitors* **	62 (45.9)	14 (37.8)	0.386	55 (30.0)	6 (40.0)	0.421
** *Statins* **	54 (40)	16 (43.2)	0.742	23 (12.5)	0 (0)	0.146
** *Diuretics* **	28 (20.9)	5 (13.5)	0.307	7 (3.8)	12 (6.7)	0.583

**Legend:** ASA: acetylsalicylic acid; diuretics (hydrochlorotiazide, furosemide); statins (atorvastatin); ACE inhibitors (ramipril, lisinopril); Ca antagonists (amlodipine, verapamil).

**Table 3 diagnostics-12-01446-t003:** Coronary anatomy and plaques feature distribution.

Group N (%)	LAP 136 (37)	n-LAP 37 (10)	*p*	CAD-RADS ≤ 1 183 (49)	CAD-RADS < 1 with pFAI ≥ −70 HU 15 (4)	*p*
**Coronary Dominance**						
** *Right* **	106 (78)	33 (89)	0.136	161 (81)	14 (93)	0.256
** *Left* **	10 (7)	3 (8)	0.835	16 (8)	1 (6)	0.963
** *Balanced* **	20 (15)	1 (3)	0.050	21 (11)	0	0.116
**Patients with HRPS**	49 (36)	7 (19)	**0.020**	0	0	
**Overall Plaque Number**	321	49	**<0.001**	0	0	
**LAD Preference**	144 (44.8)	28 (57)	0.194	0	0	
**pFAI Analysis**						
** *Volume (mL)* **	1.686 ± 0.788	1.663 ± 0.719	0.811	1.6439.719	1.657 ± 0.719	0.878
** *Median pFAI (HU)* **	−86.750 ± 10.487	−91.219 ± 9.814	**0.021**	−90 ± 10,3	64.653 ± 7.506	**<0.0001**
** *Mean pFAI (HU)* **	−90.362 ± 8.970	−94.344 ± 8.206	**0.016**	−93.5 ± 8.5	66.326 ± 5.206	**<0.0001**

**Legend:** LM: left main; LAD: left anterior descending; LCX: left circumflex; RCA: right coronary artery; HRPS: high-risk plaque signs (positive remodeling, napkin-ring sign and spotty calcifications).

**Table 4 diagnostics-12-01446-t004:** Frequency of outcomes. (a). Frequency of outcomes according to plaque type. (b). Frequency of outcomes according to pFAI.

**(a)**						
		**Composite MACEs**			** *p* **	
		**Events (N)**		**Patients (N, %)**		
**LAP**		47		136 (36,8)	**0.04**	
**n-LAP**		2		37 (9,9)		
**High-Risk Plaque Signs ***		20		70 (18,9)	**0.03**	
**(b)**						
	**Median pFAI < −70 HU** **(N = 183)**	**Median pFAI ≥ −70 HU** **(N = 189)**	** *p* **	**Mean pFAI < −70 HU** **(N = 185)**	**Mean pFAI ≥ −70 HU** **(N = 187)**	** *p* **
**CV Mortality**	2	11	**0.001**	2	11	**0.010**
**ACS**	16	29	0.051	16	29	**0.035**
**PTCA/S**	2	9	**0.004**	2	9	**0.031**
CABG	1	5	0.108	1	5	0.097

* Positive remodeling, napkin-ring sign and spotty calcifications. **Legend:** ACS: acute coronary syndrome; CABG: coronary artery bypass graft; CNG: coronarography; CV: cardiovascular; PTCA/S: percutaneous coronary angioplasty/stenting.

**Table 5 diagnostics-12-01446-t005:** Multivariable analysis. (a). Composite outcome. (b). CV mortality.

**(a)**						
**Independent Variables**	**Coefficient**	**Std. Error**	**t**	** *p* **	**r_partial_**	**r_semipartial_**
**(Constant)**	0.07242					
**BMI**	0.006654	0.005284	1.259	0.2094	0.09005	0.08755
**Dyslipidemia**	0.01966	0.06883	0.286	0.7754	0.02051	0.01986
**Diabetes**	−0.2419	0.1003	−2.411	**0.0169**	−0.1705	0.1676
**Smoke**	0.02292	0.06385	0.359	0.7200	0.02577	0.02496
**Hypertension**	−0.03415	0.06955	−0.491	0.6239	−0.03523	0.03414
**Plaque Type (LAP = 1)**	−0.1192	0.05089	−2.342	**0.0202**	−0.1658	0.1628
**pFAI_(mean)**	−0.002275	0.003338	−0.681	0.4965	−0.04886	0.04736
**(b)**						
**Independent Variables**	**Coefficient**	**Std. Error**	**t**	** *p* **	**r_partial_**	**r_semipartial_**
**(Constant)**	0.3096					
**BMI**	−0.001087	0.001423	−0.764	0.4459	−0.05490	0.05087
**Dyslipidemia**	−0.04544	0.01873	−2.426	**0.0162**	−0.1720	0.1615
**Diabetes**	−0.06300	0.02743	−2.297	**0.0227**	−0.1631	0.1530
**CAD Familiarity**	−0.003840	0.01748	−0.220	0.8263	−0.01581	0.01463
**Smoke**	−0.03002	0.01718	−1.748	0.0821	−0.1248	0.1164
**Hypertension**	0.04946	0.01835	2.695	**0.0077**	0.1904	0.1795
**pFAI_(mean)**	0.002944	0.0009139	3.222	**0.0015**	0.2259	0.2146
**Plaque Type** **(LAP = 1)**	0.04791	0.01956	2.449	**0.0152**	0.1736	0.1631

**Legend:** BMI: body mass index; LAP: low attenuation plaque; CAD: coronary artery disease.

## Data Availability

The whole raw data and results generated for this study are available on request to the corresponding author.

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
