# Peer review of "Added Value of CCTA-Derived Features to Predict MACEs in Stable Patients Undergoing Coronary Computed Tomography"

_diagnostics, 2022, doi:10.3390/diagnostics12061446_

Round 1

Reviewer 1 Report

I have read with interest the article of Pergola and coworkers, regarding the value of CCTA-derived features to predict major cardiovascular events in stable patients undergoing coronary computed tomography. It is well-established the role of coronary computed tomography as a non-invasive approaching tool to assess plaques. The aim of this study is very interesting and with important clinical implications, considering the fact that it is possible to identify a subset of patients with increased risk, who need straightening in therapeutic management and closer follow-up even in the absence of severe coronary artery disease. I consider that these findings are very interesting and can make significant contributions to further large studies. However, I consider that the structure of the article needs to be revised, to make it easier to read.  So, I suggest to insert the figures and tables immediately after their citation in the manuscript. Also, the abbreviations must be explained from their first introduction in the abstract and in the main text, even if they are well known by the practitioners. At the same time, I recommend to emphasize much better the novelty and the original elements of the study.

Author Response

We are grateful to the Reviewer for his insightful comments on our paper. We have been able to incorporate changes to reflect most of the suggestions provided. We have highlighted the changes within the manuscript.

I have read with interest the article of Pergola and coworkers, regarding the value of CCTA-derived features to predict major cardiovascular events in stable patients undergoing coronary computed tomography. It is well-established the role of coronary computed tomography as a non-invasive approaching tool to assess plaques. The aim of this study is very interesting and with important clinical implications, considering the fact that it is possible to identify a subset of patients with increased risk, who need straightening in therapeutic management and closer follow-up even in the absence of severe coronary artery disease. I consider that these findings are very interesting and can make significant contributions to further large studies.

Reply: We are grateful to the Reviewer for the positive comments on our paper.

However, I consider that the structure of the article needs to be revised, to make it easier to read.  So, I suggest to insert the figures and tables immediately after their citation in the manuscript.

Reply: we thank the Reviewer for this suggestion, and we moved the tables and figures immediately after their citation in the manuscript.

Also, the abbreviations must be explained from their first introduction in the abstract and in the main text, even if they are well known by the practitioners.

Reply: we agree with this comment, and we added all the explanation of the abbreviations

At the same time, I recommend to emphasize much better the novelty and the original elements of the study.

Reply: we are very grateful to the Reviewer for pointing out this. We implemented and enriched the Introduction and Discussion section to empathize the novelty of our study compared to previous literature.

In addition to the above comments, we improved the description of the Methods and the Results. All spelling and grammatical errors have been corrected.

We thank again the Reviewer for taking the time to read and comment our paper.

Your Sincerely

Giulio Cabrelle, MD

Valeria Pergola, MD, FESC, FEACVI

Reviewer 2 Report

The authors made an interesting study with the aim to explore the value of CCTA-derived features to predict major cardiovascular events in stable patients undergoing coronary computed tomography. The article is interesting, but the results are not clear and are not presented in an appropriate manner. So, I suggest revising the structure of the article. I also suggest to emphasize much better the results by referring to other studies, in order to highlight the novelty of the study.

Author Response

We are grateful to the Reviewer for the time and effort that have dedicated to providing a valuable feedback on our manuscript. We have been able to incorporate changes to reflect most of the suggestions provided. We have highlighted the changes within the manuscript.

The authors made an interesting study with the aim to explore the value of CCTA-derived features to predict major cardiovascular events in stable patients undergoing coronary computed tomography.

Reply: we thank the Reviewer for the positive comment on our paper

The article is interesting, but the results are not clear and are not presented in an appropriate manner. So, I suggest revising the structure of the article.

Reply: We agree with this comment. Therefore, we have clarified better the group divisions and we moved the tables and figures soon after their citation in the article. We also enriched the “CCTA Features and pFAI Values” paragraph.  This is all incorporated in the Methods and in the Result sections.

I also suggest to emphasize much better the results by referring to other studies, in order to highlight the novelty of the study.

Reply: We agree with this point and we revised the Discussion section providing comments to the previous literature and highlighting the novelty of our study. In particular we commented on the study by Nance et al, Nadjiri et al, Gaibazzi et al, Goller et al and Yan et al.

In addition to the above comments, all spelling and grammatical errors have been corrected.

We thank again the Reviewer for the insightful comments our paper.

Your Sincerely

Giulio Cabrelle, MD

Valeria Pergola, MD, FESC, FEACVI

Reviewer 3 Report

Authors in this study reported the relationship between CCTA-derived plaque features and MACE, with results showing that with pFAI incorporated into clinical risk evaluation leading to improvement in cardiac risk stratification. Result have clinical value, however, authors need to address the following comments:

1.  Abstract: pFAI-full definition should be provided. 

2. Introduction: LAP-should represent low attenuation plaque. Similarly, HAP-high attenuation plaque. There are so many short paragraphs with only one sentence, so suggest combining them into one paragraph. Authors should avoid having only one sentence paragraph. 

3. Materials and Methods: line 84, typo error:  who underwent to CCTA from...should be who underwent CCTA from...

4. Results: what does NAPs stand for? Figure 2 shows ROC analysis, however, this was not mentioned in the Methods or statistical analysis. Thus, please describe it in the Methods. 

5.  Discussion: the first paragraph needs to be extended as your discussion of your results in comparison with others is too superficial. Authors need to expand it by emphasising some key results and how this study compares to the literature. Move the study limitations before conclusion. 

Usually no citation of references in the conclusion. 

Author Response

We are grateful to the Reviewer for giving his valuable opinion to improve our manuscript. We have been able to incorporate changes to reflect all the suggestions provided. We have highlighted the changes within the manuscript.

Authors in this study reported the relationship between CCTA-derived plaque features and MACE, with results showing that with pFAI incorporated into clinical risk evaluation leading to improvement in cardiac risk stratification. Result have clinical value, however, authors need to address the following comments:

  1. Abstract: pFAI-full definition should be provided. 

Reply: we agree with this comment, and we incorporated the full definition of pFAI in the Abstract section (line 33).

  1. Introduction: LAP-should represent low attenuation plaque. Similarly, HAP-high attenuation plaque. There are so many short paragraphs with only one sentence, so suggest combining them into one paragraph. Authors should avoid having only one sentence paragraph. 

Reply: We thank the Reviewer for this suggestion. We changed the definition of LAP and HAP (line 63) and we combined the sentences in one paragraph as suggested (lines 65-76).

  1. Materials and Methods: line 84, typo error:  who underwent to CCTA from...should be who underwent CCTA from...

Reply: we changed the typo error (line 94), thanks for pointing out.

  1. Results: what does NAPs stand for? Figure 2 shows ROC analysis, however, this was not mentioned in the Methods or statistical analysis. Thus, please describe it in the Methods. 

Reply: We are extremely sorry, NAPs is another typo mistake, it stands for n-LAPs, we changed it. We included the description of ROC analysis in the Statistical analysis section (lines 181-183)

  1. Discussion: the first paragraph needs to be extended as your discussion of your results in comparison with others is too superficial. Authors need to expand it by emphasizing some key results and how this study compares to the literature. Move the study limitations before conclusion. 

Reply: we have, accordingly, revised the Introduction and the Discussion section, comparing our results to previous studies emphasizing the novelty of our paper. In particular we discussed the papers by Nance et al, Nadjiri et al, Gaibazzi et al, Goller et al and Yan et al.

Usually no citation of references in the conclusion. 

We agree with this comment, and we removed the citation from the Conclusion section.

We thank again the Reviewer for providing valuable suggestion and comments to our paper.

Your Sincerely

Giulio Cabrelle, MD

Valeria Pergola, MD, FESC, FEACVI

Round 2

Reviewer 3 Report

Thank you for your effort in revising the manuscript and addressing my comments. I do not have any further comments.